# Dynamic Capabilities Influence on the Operational Performance of Hotel Food Supply Chains: A Mediation-Moderation Model

Mahmoud Abou Kamar [1] , Omaima Munawar Albadry [2] , Samar Sheikhelsouk [3], Mohammed Hasan Ali Al-Abyadh [4] and Omar Alsetoohy [1,*]

1   Faculty of Tourism and Hotels, University of Sadat City, Sadat City 32897, Egypt; mahmoud.aboukamar@fth.usc.edu.eg
2   College of Business Administration, Jazan University, Jazan 45142, Saudi Arabia; oalbadry@jazanu.edu.sa
3   Faculty of Business, Menofia University, Menofia 32511, Egypt; samarm000@commerce.menofia.edu.eg
4   College of Education in Wadi Alddawasir, Prince Sattam bin Abdulaziz University, Al-Kharj 16278, Saudi Arabia; m.alabyadh@psau.edu.sa
*   Correspondence: omar.alsetoohy@fth.usc.edu.eg

**Abstract:** This study develops and tests an integrated model based on the Dynamic Capabilities View (DCV) to empirically examine how dynamic capabilities influence the operational performance of hotel food supply chains through the mediating role of supply chain resilience and the moderating influences of environmental uncertainty and disruption orientation. The model is tested using survey data from 160 five- and four-star hotel managers in Egypt and the findings of structural equation modeling. The findings support the proposed model and reveal a positive effect of total dynamic capabilities and the four dynamic capabilities (i.e., collaboration, integration, agility, and reconfiguration) on the operational performance of hotel food supply chains through the mediating role of supply chain resilience. The results affirm that supply chain resilience mediates the relationship between dynamic capabilities (in total) and operational performance. Furthermore, the results show that environmental uncertainty moderates the above linkage, whereas disruption orientation does not do that. With the extension of DCV, our findings contribute to deepening our understanding of the dynamic capabilities contributing to the development of hotel food supply chain performance. These findings hold crucial implications for academics, managers, and policymakers. They also provide valuable insights on how to effectively control operational performance during disruptions.

**Keywords:** hotel food supply chain; dynamic capabilities; resilience; operational performance; Egypt

## 1. Introduction

Globally, food supply chains are facing difficult times of volatility and uncertainty. Not even a few months after recovering from the devastating effects of the COVID-19 pandemic, food supply chains have been hit with a new disruption: the Russian-Ukrainian conflict and the loss of Ukrainian exports. These unexpected events have disrupted the food supply chains and increased oil prices, increasing transportation and manufacturing costs [1].

Hotel food supply chains (FSCs) are more vulnerable to such disruptions [2], which lead to capacity gaps [3] and financial loss [4]. The key challenge for hotel FSCs lies in how to orchestrate the flow of resources while improving the resilience of supply systems to recover more effectively from disruptions [2]. In the current scenario, it is no longer enough to mitigate the risks. Most importantly, hotel FSCs must develop the dynamic capabilities needed to adapt to the changing business environment, which Hussain & Malik [5] have defined as a firm's ability to detect threats and opportunities in the market, seize beneficial market opportunities, and then change its existing resource base to effectively traverse

volatility in the market. For instance, due to the high demand for local foods among customers since the COVID-19 outbreak, conventional food providers (i.e., restaurants, hotels, food stores, etc.) have adapted and shifted their food supply chains to use local, organic, and sustainable local foods in their menus [6].

Consequently, this has prompted us to respond to recent calls in the literature [5,7–9] and to draw on the dynamic capabilities view (DCV) to investigate the impact of dynamic capabilities on the operational performance of hotel FSCs through resilience. Traditionally, DCV [10] extends the resource-based view by focusing on a firm's ability to compete in dynamic markets characterized by rapid and unpredictable changes [11]. DCV has been thoroughly explored in supply chain management [9,12]. These studies conclude that dynamic capabilities can be established in collaboration with other partners to increase supply chain effectiveness [13], competitiveness [14], sustainability [15], and performance [16]. In the wake of recent turbulence in the hospitality industry, there has been a resurgence of interest in dynamic capabilities because of their potential to alter "how a firm earns its existing strength" to adapt to shifting market conditions [17].

DCV has been criticized for its lack of "empirical foundations" [11]. Existing studies that adopt the dynamic capabilities paradigm fail to identify procedures, resources, and pathways that boost supply chain competencies [18]. Supply chain resilience is widely recognized as an essential supply chain capability to manage unforeseen circumstances. This capability can mitigate the susceptibility of organizations and facilitate their resilience in the face of disruptions, enabling them to recover their previous operational condition, if not surpass it [19]. Supply chain resilience in enterprises contributes to decreasing vulnerability and ensuring seamless operations [20].

Similarly, Chowdhury and Quaddus [18] have argued that to tackle the obstacles that come along with unpredictable and constantly changing conditions, companies must adopt a resilient strategy for their supply chains. This can be achieved by establishing a supply chain disruption focus, ensuring resource configuration, and implementing a robust risk management framework. El-Baz & Ruel [21] distinguish between the robustness and resilience of supply chain management. The former refers to the ability of a supply chain to maintain its planned level of performance in the face of disruptions. In contrast, the latter pertains to the ability of the supply chain to recover its performance following the absorption of disruptive effects. Thus, organizations have directed considerable investments towards the development of capabilities to mitigate the deleterious effects of supply chain disruptions, with the ultimate goal of enhancing supply chain performance. Recently, organizations have developed dynamic capabilities in response to intense competitive pressures and dynamic surroundings by integrating, constructing, and reconfiguring internal and external competencies [22]. The effects of informational, relational, and integration capabilities on supply chain resilience have been observed and documented in practical and theoretical research studies. For instance, Brusset and Teller [23] and Liu and Lee [24] argue that integration capability has significantly affected supply chain resilience. In addition, several recent studies on supply chains [18,25,26] have applied DCV to investigate the relationships between dynamic capabilities, turbulent business environments, internal environments, competitiveness, and financial performance. The existing body of literature mainly focuses on analyzing traditional supply chain practices within a stable business context. Little attention has been devoted to studying the effects of dynamic capabilities on the resilience of hotel FSCs during disruptions, such as the COVID-19 pandemic [3]. According to Queiroz et al. [27], coping, recovery, and resilience dynamic capabilities need further study. Prior research studies, e.g., [18,20,28], confirm the need for more research on the role of supply chain dynamic capabilities in enhancing supply chain performance after disruptions. Additionally, Shen & Sun [29] have stated that future research should focus on the linkage between supply chain resilience and the capabilities of the enterprise to enhance performance.

Similarly, Sousa & Voss [30] have discussed that it is important for researchers to consider the environmental factors that contribute to the effectiveness of a capability,

especially when there is empirical evidence to support its performance value. Moreover, the literature has revealed that environmental uncertainty, specifically market turbulence and regulatory uncertainty, affects supply chain resilience and efficiency [22,31]. Havakhor et al. [32] have asserted that firms face challenges with regard to decision-making related to capability-building targets due to increased environmental uncertainty, which leads to suboptimal utilization of these capabilities. Thus, to achieve supply chain resilience, which, in turn, results in better supply chain performance, companies should consider various channel initiatives based on environmental uncertainty levels [31].

Hence, considering the aforementioned gaps, this study contributes to filling these research gaps by developing an integrated model based on DCV to investigate the influence of dynamic capabilities of hotel FSCs (i.e., collaboration, integration, agility, responsiveness, and reconfiguration) on resilience and hotel performance. This is done while considering the moderating role of environmental uncertainty and disruption orientation.

The pursuit of these objectives leads to three major contributions. First, it is significantly important for hotels to manage the pace of changes in both products and processes, coinciding with disruptions since they affect supply chain partners [33]. Due to the complexity of hotel FSCs, any single activity in the chain carries an inherent risk that may result in unforeseen problems at other stages and cause financial losses [3,6,34]. Teece [35] and Teece et al. [36] have pointed out that the relationship between the firm and the environment in which it operates is essentially symbiotic. Therefore, it is important to understand how dynamic capabilities build the resilience of hotel FSCs through a range of proactive, collaborative, and reactive capabilities that are appropriate to the hotel business and are set to achieve higher operational performance.

Second, the importance of supply chain resilience is that it enables hotels to better manage disruptions and, as a result, maintain their operational performance [23,37]. Thus, this study examines the relationship between hotel FSCs resilience and its implications for operational performance during disruptions.

Third, we seek to understand the moderating roles of environmental uncertainty and disruption orientation with regard to the relationships between hotel supply chain resilience and operational performance. The results should motivate hotel operators and decision-makers to become more proactive and take initiatives to boost the efficiency of their food supply chains.

## 2. Theoretical Background and Hypotheses Development

This study draws upon DCV to propose an integrated model (see Figure 1) and empirically tests the relationships among dynamic capabilities, supply chain resilience, environmental uncertainty, disruption orientation, and supply chain operational performance.

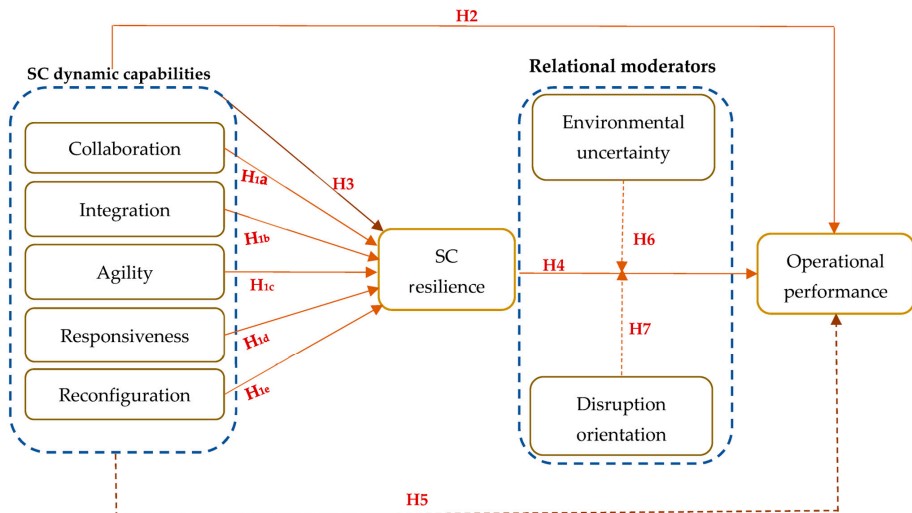

**Figure 1.** The Theoretical Framework and Hypotheses.

*2.1. Dynamic Capabilities View (DCV)*

The dynamic capability view, according to Teece et al. [10], is based on the resource-based view [38]. DCV, as Teece [35] outlined, is an organizational process that allows businesses to detect risks, seize opportunities, and maintain competitiveness by using technological advancements, mergers, protections, and reorganizations of both intangible and tangible assets. Implementing DCV in the supply chain is gaining traction. Oh et al. [39] define dynamic supply chain capabilities as the ability of a firm to effectively and efficiently utilize both internal and external resources to enhance supply chain practices and improve performance. These capabilities encompass sharing information, coordination, integration, and supply chain responsiveness. Furthermore, according to Ju et al. [40], dynamic supply chain capabilities encompass various processes, including information exchange, supply chain alignment, and information technology, to effectively meet customer demands and maintain competitiveness in a rapidly changing environment. Additionally, Aslam et al. [41] emphasize the significance of supply chain agility and adaptability as crucial components of dynamic supply chains.

Several studies [3,35,40,42] suggest that dynamic capabilities are considered high-order capabilities, which may be further broken down into various capacities. To date, there is no globally accepted categorization of the dynamic capabilities of hotel supply chains. Therefore, we have disaggregated the dynamic capabilities of supply chains into five distinct categories: collaboration, integration, agility, responsiveness, and reconfiguration, based on a thorough review of the literature. Each sub-capabilities has been carefully chosen to reflect hotel FSCs' ability to adapt and promptly address the ever-changing market dynamics. For instance, the concept of supply chain collaboration capability pertains to a hotel's capacity to foster enduring partnerships involving various activities and exchange information, resources, and risks to attain common goals [43]. Yunus [44] emphasizes the importance of customer collaboration, supplier collaboration, and internal collaboration as essential elements of a collaborative supply chain. Moreover, the integration capability pertains to its proficiency in establishing strategic partnerships and collaborating effectively with supply chain partners [45], which encompasses the seamless integration of information, physical, and financial flow [46]. Angeles [47] has argued that the main objective of supply chain integration is to ensure that customers have access to the right products at the right time at a competitive price.

Additionally, the agility capability of hotel FSCs is related to their capacity to promptly adapt to market changes and uncertainties for the benefit of their suppliers and customers [41]. Furthermore, it is a dynamic process that involves modifying or reorganizing business operations to address market fluctuations and other uncertainties. Kareem & Kummitha [42] suggest that supply chain agility encompasses essential components such as strategic, operational, and episodic preparedness and responsiveness. Hotel FSCs responsiveness refers to the partners' ability to effectively adapt to changes and fluctuations in the environment to reduce lead time, enhance service quality, quickly meet customer demands, and optimize transportation [48]. Supply chain responsiveness includes three essential components: agility in meeting customer requirements, flexibility in supporting new product development and market entry, and mitigating the risk of supply chain bottlenecks and disruptions [42]. Finally, supply chain reconfiguration involves making structural and functional adjustments that may be prompted by a disruptive event to enhance the supply chain. Reconfiguration is typically necessary following a supply chain failure. However, it can be utilized as an innovation strategy to improve performance. Each level within the supply chain has specific parameters and attributes that determine the extent of reconfigurability, enabling the selection of the most appropriate configuration [49].

2.1.1. Collaboration as a Dynamic Capability

Supply chain collaboration allows synergistic partnerships in relation to processes such as forecasting and risk management models [50], which are essential for reducing the level of uncertainty and mitigating potential risks through the exchange of information [51].

The cornerstone of the collaboration is achieving mutual benefits and sharing common risks [52]. According to Dougherty et al. [53], supply chain collaboration includes information exchange, development of common strategic goals, and synchronization of operations. Simatupang & Sridharan [54] have emphasized that information sharing and decision synchronization are the main pillars of collaboration. In risky event scenarios, collaborative activities increase supply chain resilience because they help firms avoid overreactions, unnecessary interventions, and fruitless judgments [55].

Moreover, collaboration encourages innovation in problem-solving to address new issues as they crop up [56]. Following previous studies [9,57,58], the current study can argue that collaboration activities may positively enhance the resilience of the food supply chain. Hence, we hypothesize the following:

**H1a.** *The higher level of collaboration positively influences the resilience of hotel FSCs.*

### 2.1.2. Integration as a Dynamic Capability

Integration with partners includes two primary paths: resource planning and operations, such as inventory management [23]. These two ways of integrating can make supply chains more resilient by encouraging a continuous and powerful flow of products, services, information, money, and decision-making elements to deliver maximum value and efficiency with the minimum expense [59]. Integration seeks to streamline processes across the supply chain to enhance resilience and performance by ensuring product quality and diversity [60]. Supply chain integration may also include strategic decisions to facilitate the exchange of important information regarding new markets, goods, consumers, and future markets. In the context of Industry 4.0, the integration of the supply chain has three parts: the integration of processes and activities, the integration of technologies and systems, and the integration of organizational relationships [61]. Recent studies have confirmed the positive relationship between integration capability and supply chain resilience [23,60,62]. Hence, we submit the following hypothesis:

**H1b.** *The higher level of integration positively influences the resilience of hotel FSCs.*

### 2.1.3. Agility as a Dynamic Capability

Agility has recently emerged in the context of dynamic capabilities that support supply chains to thrive in unpredictable marketplaces [63,64]. Typically, agility is the ability of the supply chain to swiftly change strategies and procedures in response to environmental uncertainties. The resulting capability can be applied proactively or reactively to develop a superior competitive position by swiftly reacting to market volatility [65]. In a dynamic corporate environment, "it is not the large that devours the small; it is the swift that consumes the slow" [64]. Supply chain agility can reduce the likelihood of supply chain disruptions by enabling firms to sense environmental threats [35] and respond to them using resource reconfiguration, collaborative supplier networks, and collaborative infrastructure [64]. Several empirical studies have shown that supply chain resilience can be improved with more agility [65,66]. Therefore, we propose the following:

**H1c.** *The higher level of agility positively influences the resilience of hotel FSCs.*

### 2.1.4. Responsiveness as a Dynamic Capability

Responsiveness is the ability to promptly and systematically respond to volatility and vulnerability in the business environment [67]. Many businesses now recognize the importance of supply chain responsiveness as an important capability to possess [68]. Thus, responsive supply chains are essential to a firm's survival and long-term success in the face of rising competition and shifting customer demands [69]. The findings of a recent empirical study have indicated that despite the severe disruptions in the business environment, improving supply chain responsiveness in times of crisis has contributed to mitigating negative impacts and enhancing the resilience of supply chains since it has enabled the skipping of non-essential tasks that take a long time and eased bottlenecks.

It also facilitates better allocation and prioritization [70]. Besides, Munir et al. [71] have confirmed the positive relationship between responsiveness and SC resilience. This leads us to suggest the following hypothesis:

**H1d.** *The higher level of responsiveness positively influences the resilience of hotel FSCs.*

2.1.5. Reconfiguration as a Dynamic Capability

Firms' survival depends on their ability to manage and reconfigure resources during disruptions [72]. The high unpredictability surrounding supply chain disruptions raises questions regarding the worth of the current resources in generating capabilities to recover from disruption. To sense threats and seize opportunities, firms may need to reconfigure their scarce resources to adapt to turbulent and unpredictable environments [73]. Studies show that in times of crisis, resource reconfiguration is critical to the survival of the supply chain [72]. Firms that have experienced dealing with disruptions are more likely to set up, align their resources, and give themselves enough time to scan the environment to figure out how to respond to a potential disruption [27]. As a result, we suggest the following hypothesis:

**H1e.** *Resource reconfiguration positively influences the resilience of hotel FSCs.*

2.1.6. Dynamic Capabilities, Supply Chain Resilience, and Operational Performance Improvement

The hospitality sector is a highly dynamic environment; hence, hotels must possess supply chain dynamic capabilities to adapt and respond to changes to sustain and achieve better performance [74,75]. In a recent study, Zhao et al. [20] argue that multiple studies have shown the positive impact of dynamic capabilities on supply chain performance. For instance, Yook et al. [76] confirm the significant influence of dynamic capabilities on economic and environmental performance. Similarly, Kareem & Kummitha [42] have mentioned that supply chain dynamic capabilities are positively correlated to the operational performance of manufacturing companies in Hungary. Additionally, it is found that supply chain sustainability management (SSCM) practices can enhance dynamic capabilities, leading to significant improvements in environmental performance [77]. Rauer & Kaufmann [78] have delved into the identification of dynamic capabilities that can be employed to overcome barriers to green supply chain management.

On the other hand, a resilient supply chain can endure change, adapt to disruption, and improve operational performance [79]. Resilience is the ability of a supply chain to react to and recover from unexpected events [80]. According to Ivanov [81], resilience is an active part of operational management decisions that create value. Supply chain resilience has recently been found to improve financial performance [3] and organizational and operational performance [82]. From a dynamic perspective, supply chain resilience reduces operational disruptions and allows firms to improve operations. Thus, there will be fewer pauses in product deliveries and fewer cash flow problems.

According to the conceptual definition of resilience, it is deemed suitable to consider the time required for recovery and resuming normal operations after a disruption as a quantitative resilience measure [83]. Additionally, the available literature proposes two other metrics related to recovery for assessing resilience: one that evaluates the level of recovery achieved after recovery periods and a second that measures the loss in performance experienced by the supply chain during recovery periods [84].

To establish a formal framework for measuring resilience, it would be beneficial to consider utilizing operational performance to evaluate the effectiveness of resilient solutions. Operational performance pertains to a hotel's capability to decrease management expenses and lead times while enhancing the utilization of resources and distribution capacity [85]. Operational performance holds significant value for hotel FSCs as it is directly related to production efficiency and creating top-notch products, ultimately resulting in amplified profitability and competitiveness [77]. The successful translation of operational capabilities into competitive advantages for firms is one feature of the multi-faceted concept

known as operational performance. Productivity, quality, pricing, delivery, and adaptability are some ways to measure it [42]. Supply chain resilience is found to have a positive impact on firms' operational performance [86]. Chowdhury et al. [87] prove a correlation between supply chain resilience and supply chain performance in manufacturing firms. According to recent research conducted by Alkhatib et al. [88], there is a notable and favorable relationship between the adoption of supply chain resilience practices and overall operational performance in Jordanian manufacturing firms. This is in addition to the confirmed significant influences of dynamic capabilities on supply chain resilience [23,24]. Alkalha et al. [89] have stated that supply chain resilience strongly mediates the relationship between dynamic capabilities in terms of absorptive capacity and operational performance. Hence, the following hypotheses were put forward:

**H2.** *Hotels' dynamic capabilities, in all, are positively related to operational performance;*

**H3.** *Hotels' dynamic capabilities, in all, are positively related to food supply chain resilience;*

**H4.** *Hotel food supply chain resilience is positively related to operational performance;*

**H5.** *Food supply chain resilience positively mediates the relationship between hotels' dynamic capabilities and operational performance.*

### 2.2. The Moderating Role of Environmental Uncertainty and Disruption Orientation

Hotel FSCs face many risks; perhaps the most prominent is uncertainty [36]. Uncertainty becomes an issue when it interacts with a firm's critical features and affects its efficacy [90]. Several studies on dynamic capabilities identify environmental uncertainty as a key factor in market dynamism [36,91]. According to Teece [35], dynamic capabilities are important in dynamic settings. By definition, a dynamic business environment is always evolving due to unanticipated changes in the market [91]. With this dilemma, hotels have no choice but to use dynamic capabilities to achieve targeted operational performance [92]. In times of uncertainty, environmental analysis could help hotels achieve better supply chain resilience and operational performance [75]. In this study, environmental uncertainty is considered a moderator of supply chain resilience and operational performance. Hence, we suggest the following:

**H6.** *Environmental uncertainty moderates the relationship between supply chain resilience and operational performance.*

Similarly, disruption orientation is an essential factor in the reconfiguration of dynamic capabilities in a highly disruptive environment [93,94]. The concept is related to a firm's attempts to continuously monitor the environment to anticipate risks and proactively learn from past events to mitigate disruptions and seize opportunities [3]. Several recent studies [93,95,96] have shown that disruption orientation helps firms enhance their supply chain performance by investing in knowledge management to monitor their supply chains and acquire and analyze real-time data using digital twin technology. Therefore, it could be argued that:

**H7.** *The more flexible the disruption orientation, the stronger the relationship between supply chain resilience and operational performance.*

### 3. Methodology

#### 3.1. Survey Administration and Data Collection

Since we are investigating an integrated model driven by an established theory (DCV), we have developed a structured questionnaire to collect data from hotels in Egypt. The five- and four-star hotels operating in Greater Cairo, the Northern Coast, Sharm El Sheikh, Dahab, El Gouna, Marsa Alam, Hurghada, and Ras Sidr from 2021 to 2022 were the focus of this study. These destinations were chosen because they represent the major geographical areas in Egypt, including the largest number of four and five-star hotels, and comprise diverse and complex food supply chains. According to the Egyptian Hotel Directory for

2022, there are 171 four-star hotels and 106 five-star hotels in these destinations. Electronic emails were collected from the hotels' websites, and requests to participate in the survey were sent out. Decision-makers in supply chain management or operation management-related jobs were the targeted respondents. Initially, an electronic invitation including the purpose of the study was forwarded to the selected hotels.

Furthermore, they defined the important terms and acronyms used, survey questions, and a pledge to keep the responses confidential. A gentle reminder was floated between 1 February and 30 March 2023. After several waves of follow-ups with non-responders, all the data were collected by April 2023. Out of the 277 surveys received, 160 responses were qualified: 85 surveys from four-star hotels and 75 surveys from five-star hotels, with a response rate of 58%. The response rate has been deemed exceptional [97,98]. For ascertaining the adequacy of the final sample, an online calculator designed by Soper [99] was utilized. The model parameters of the study consisted of nine theoretical constructs, which were measured using 46 survey questions. A 0.5 effect size was assumed, with a desired statistical level of 0.80 and a probability for Type-1 error set at 0.05. Based on these parameters, a minimum sample size of 107 was recommended. However, the final sample size 160 surpasses this recommendation, indicating sufficient adequacy. Thus, the sample size has been considered suitable for further analysis.

### 3.2. Instrument and Measurement

To create effective measures, the study attempted to define the specification of each construct, which was endorsed by the collection and analysis of measurement items from previous studies. With the support of an expert panel of both academics and industry professionals, the content validity of the measurement scales was further established to increase the accuracy of the study. Following the comments and observations of the expert panel, the metric formulations were modified (see Appendix A). Twenty-six hotels were randomly selected from the Egyptian Hotel Directory for 2022, and data were collected for a pilot study. To ensure that the study constructs are unidimensional, an exploratory factor analysis was performed. Items with loads greater than 0.5 were classified as capable of scaling the intended structures. The development of sufficient factor loadings allowed all elements of the original study to be retained. After the pilot test, the instrument was ready to collect the final data.

In addition, the survey comprised two sections. Validated scales borrowed from previous studies [64,100–102] formed the Section 1. Dynamic capabilities were measured using the five domains [independent variables] of collaboration (5 items), integration (8 items), agility (5 items), responsiveness (5 items), and reconfiguration (3 items). Six metrics modified from Ambulkar et al. [103] were used to assess the hotels' FSCs resilience. There were also two moderators incorporated into the model: environmental uncertainty and disruption orientation. As Ambulkar et al. [103] indicated, six items were adopted for environmental uncertainty. On the other hand, four items for disruption orientation were appropriated from Aslam et al. [41]. Finally, to examine operational performance as a dependent variable, six measures were adopted from Bag & Rahman [104] and Siagian et al. [60]. On a five-point Likert scale, where "strongly disagree" = 1 and "strongly agree" = 5, respondents were asked to rate their level of agreement or disagreement. Section 2 of the survey was dedicated to a breakdown of the respondents' characteristics (see Table 1).

Table 1 shows that 48% of respondents are chain or affiliated groups, whereas 46% are locally owned and operated. Moreover, 37% of the hotels have been in business for 10–15 years, and 36% have been operating for 15–20 years. As for average annual earnings (EGP million), 48.7% of the hotels obtain average profits of 100 million to less than 300 million, while 48.1% have achieved average profits of more than 300 million. 33% of hotels have 3–6 food and beverage outlets, while 36% have 6–9. Finally, 42.5% of hotels have collaborated with suppliers for 3–6 years, 35% for less than three years, and 15% for 6–9 years.

**Table 1.** Characteristics of the Respondents (n = 160).

| Attribute | Five-Star Hotels (n = 75) | | Four-Star Hotels (n = 85) | |
|---|---|---|---|---|
| | **Freq.** | **%** | **Freq.** | **%** |
| **Property ownership structure** | | | | |
| Chain/group affiliation | 38 | 50.7 | 39 | 45.9 |
| Independently locally owned/operated | 28 | 37.3 | 45 | 53.0 |
| Wholly foreign-owned operation | 9 | 12.0 | 1 | 1.2 |
| **Duration of market presence** | | | | |
| Less than 10 years | 16 | 21.3 | 16 | 18.8 |
| 10–15 years | 29 | 38.7 | 30 | 35.2 |
| 15–20 years | 19 | 25.3 | 38 | 44.7 |
| >20 year | 11 | 14.7 | 1 | 1.2 |
| **Annual profits (NT) last year (EGP Million)** | | | | |
| <100 million | 0 | 0 | 5 | 5.9 |
| From 100 to less than 300 million | 36 | 48.0 | 42 | 49.4 |
| >300 million | 39 | 52.0 | 38 | 44.7 |
| **The total number of F&B operations** | | | | |
| <3 outlets | 0 | 0 | 4 | 4.7 |
| 3–6 outlets | 25 | 33.3 | 28 | 33.0 |
| 6–9 outlets | 27 | 36.0 | 31 | 36.5 |
| >9 outlets | 23 | 30.7 | 22 | 25.8 |
| **The number of hotel food suppliers** | | | | |
| <20 suppliers | 28 | 37.3 | 50 | 58.8 |
| From 20 to 50 suppliers | 29 | 38.7 | 34 | 40.0 |
| >50 suppliers | 18 | 24.0 | 1 | 1.2 |
| **Collaboration time with suppliers** | | | | |
| Less than 3 years | 22 | 29.3 | 34 | 40.0 |
| 3–6 years | 34 | 45.3 | 34 | 40.0 |
| 6–9 years | 13 | 17.3 | 11 | 12.9 |
| 9–12 years | 6 | 8.0 | 6 | 7.6 |
| >12 years | 0 | 0 | 0 | 0 |

### 3.3. Data Processing and Analysis

The partial least squares-structural equation modeling (PLS-SEM) approach was used to test the study's hypothetical model. This approach allows us to test sophisticated models, such as in the case of the current study, where there were no technical restrictions. PLS-SEM uses path analysis, regression analysis, and confirmatory factor analysis to verify model validity [105]. Later, indicator loadings were assessed; as a result, some items were eliminated (Appendix A). The loading of other items exceeded 0.7 [106], except for (AGI1, AGI4, and DIO1) greater than 0.55. Cronbach's alpha and composite reliability (CR) were also used to verify the reliability of the constructs, while convergent and discriminant validity were used to confirm the model's validity [105]. The results of these tests are presented in Table 2. For the first test, the average variance extracted (AVE) was employed, and the square roots of the AVEs were utilized (see Table 3). In Table 2, both the composite reliability (CR) and Cronbach's alpha values seem to be higher than 0.7, which is higher than the sufficient threshold for the reliability of latent variables [105]. This shows that all the constructions are internally consistent, indicating strong composite reliability.

**Table 2.** Measurement Model: Quality Criteria of the Constructs.

| Constructs | CR | Cronbach's $\alpha$ | AVE |
|---|---|---|---|
| Collaboration capability (COL) | 0.91 | 0.88 | 0.68 |
| Integration capability (INT) | 0.93 | 0.91 | 0.69 |
| Agility capability (AGI) | 0.85 | 0.77 | 0.53 |
| Responsiveness capability (RES) | 0.92 | 0.88 | 0.80 |
| Reconfiguration capability (REC) | 0.92 | 0.87 | 0.79 |
| SC Resilience (SCR) | 0.96 | 0.95 | 0.79 |
| Environmental uncertainty (ENU) | 0.95 | 0.93 | 0.75 |
| Disruption orientation (DIO) | 0.88 | 0.83 | 0.65 |
| Operational Performance (OPP) | 0.90 | 0.85 | 0.70 |

CR = Composite Reliability, $\alpha$ = Cronbach's alpha, AVE = Variance Extracted.

**Table 3.** Discriminant Validity and Correlation between Latent Variables (Fornell & Larcker Criteria) [107].

| | AGI | COL | DIO | ENU | INT | OPP | REC | RES | SCR |
|---|---|---|---|---|---|---|---|---|---|
| AGI | **0.73** | | | | | | | | |
| COL | 0.72 | **0.83** | | | | | | | |
| DIO | 0.68 | 0.59 | **0.81** | | | | | | |
| ENU | 0.78 | 0.56 | 0.84 | **0.86** | | | | | |
| INT | 0.67 | 0.73 | 0.57 | 0.48 | **0.83** | | | | |
| OPP | 0.50 | 0.82 | 0.28 | 0.25 | 0.66 | **0.83** | | | |
| REC | 0.86 | 0.71 | 0.70 | 0.79 | 0.62 | 0.44 | **0.89** | | |
| RES | 0.88 | 0.81 | 0.65 | 0.76 | 0.59 | 0.54 | 0.86 | **0.89** | |
| SCR | 0.86 | 0.86 | 0.72 | 0.77 | 0.77 | 0.65 | 0.88 | 0.85 | **0.89** |

Bold values represent the squared root estimate of AVE.

The data in Tables 2 and 3 demonstrate that all the instruments included in the study have adequate convergent and discriminant validity. In this regard, the AVE of all constructs is higher than the convergent validity criterion of 0.5, which is the threshold value suggested by Hair et al. [105]. The approach developed by Fornell & Larcker [107] evaluates discriminant validity (AVE). They have proposed that discriminant validity might be supported if the square root of the AVE (diagonal element) for a latent variable is larger than the correlation values among all latent variables (absolute values of off-diagonal elements). This would indicate that AVE is sufficiently discriminatory.

Furthermore, the data in Table 3 show that AVEs are greater than other correlations, including the inter-construct correlations, thereby confirming discriminant validity [108]. There are no issues with multicollinearity or common method bias because none of the VIF values is higher than 5; all these values are lower than 5 [109]. In addition, evidence of discriminant validity is shown when the average variance extracted (AVE) for a latent variable is greater than the maximum shared variance (MSV) with other latent variables [105]. These results show that our measurement model is well-defined, where the constructs are valid and highly reliable.

## 4. The Structural Model and Hypotheses Testing Results

SEM with SmartPLS V.3 was operationalized to analyze the PLS path modeling and test the study hypotheses. The values of the hypothetical model's path coefficients ($\beta$), *p* values, $R^2$, and $Q^2$ are determined. The findings of the test of the hypothesis are clarified in Table 4, as well as in Figures 2 and 3.

**Table 4.** Results of the Hypotheses Testing.

| No. | Hypotheses | B | *p*-Value | Result |
|---|---|---|---|---|
| H1a | Collaboration capability -> SC resilience | 0.37 *** | 0.001 | supported |
| H1b | Integration capability -> SC resilience | 0.15 *** | 0.001 | supported |
| H1c | Agility capability -> SC resilience | 0.19 * | 0.02 | supported |
| H1d | Responsiveness capability -> SC resilience | −0.080 | 0.33 | Not supported |
| H1e | Reconfiguration capability -> SC resilience | 0.44 *** | 0.001 | supported |
| H2 | Hotel dynamic capabilities (in total) -> Operational performance | 0.550 *** | 0.001 | Supported |
| H3 | Hotel dynamic capabilities (in total) -> SC resilience | 0.94 *** | 0.001 | supported |
| H4 | SC resilience -> Operational performance | 0.40 *** | 0.00 | supported |
| H5 | SC resilience -> mediates SC dynamic capabilities and operational performance | 0.380 *** | 0.001 | Supported |
| H6 | Environmental uncertainty -> moderates SC resilience and operational performance | −0.38 * | 0.012 | supported |
| H7 | Disruption orientation -> moderates SC resilience and operational performance | 0.10 | 0.50 | Not supported |

\* Significance at 1%, \*\*\* Significance at 0.1%.

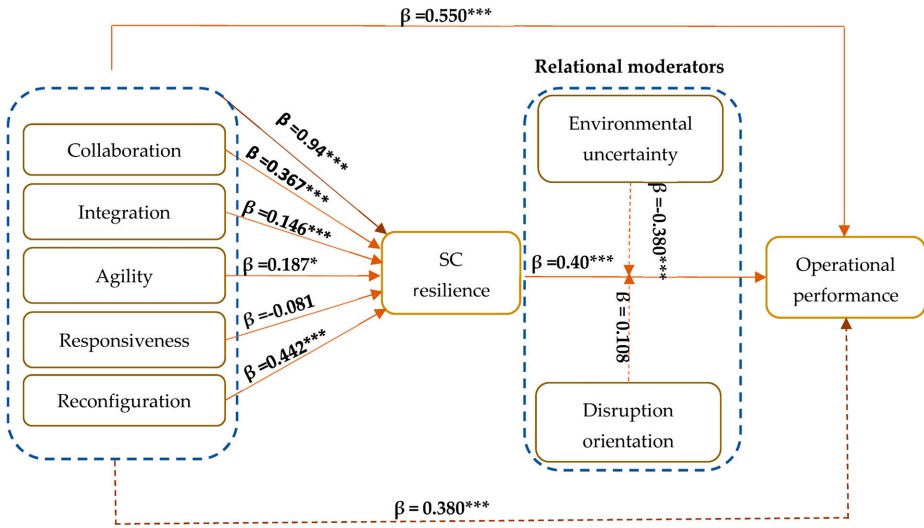

**Figure 2.** The Results of the Research Framework. * Significance at 1%, *** Significance at 0.1%.

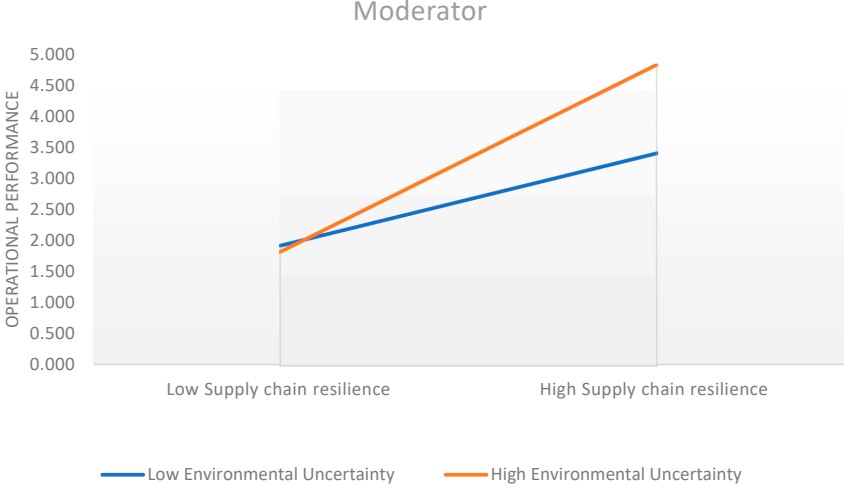

**Figure 3.** The Moderating Effect of Environmental Uncertainty.

*4.1. Direct Relationship Results*

Of the various dynamic supply chain capabilities, only four (i.e., collaboration, integration, agility, and reconfiguration) have had the most significant impact on supply chain resilience, with a corresponding value of ($\beta = 0.367$, $p < 0.001$; $\beta = 0.146$, $p < 0.001$; $\beta = 0.187$, $p < 0.05$; and $\beta = 0.442$, $p < 0.001$), respectively. As a result, hypotheses H1a, H1b, H1c, and H1e are supported. Conversely, H1d has not been supported due to the insignificant relationship between responsiveness capability and SC resilience ($\beta = -0.08$, $p > 0.05$). Additionally, the hotel's dynamic capabilities (in total) are positively correlated with the operational performance ($\beta = 0.550$, $p < 0.001$), supporting H2. In the same vein, the results show a strong positive relationship between the hotel's dynamic capabilities (in total) and SC resilience ($\beta = 0.94$, $p < 0.001$), supporting H3. Moreover, the findings indicate that the resilience of the supply chain has a significant positive effect on operational performance ($\beta = 0.40$, $p < 0.001$), which lends credence to hypothesis H4. This clarifies that supply chain resilience improves the overall operational effectiveness of hotel FSCs.

*4.2. Indirect Relationship Results*

4.2.1. The Mediation Results

To test the mediation effects, we have followed the proposed approach for mediation by Kock [110], which improves upon the methods introduced by Preacher & Hayes [111] and Hayes & Preacher [112] by offering greater efficiency and reliability. Our study revealed a direct and significant correlation between the hotels' dynamic capabilities and operational performance and between the hotels' dynamic capabilities and SC resilience. According to our findings, the cumulative sum of the indirect effects of a hotel's dynamic capabilities on operational performance, which encompasses only one path (i.e., hotel's dynamic capabilities -> SC resilience -> operational performance), is found to be highly significant ($\beta = 0.94$, $p < 0.001$, f2 = 0.332), with a large effect size [113]. Moreover, the bootstrapped confidence interval (LL = 0.412, UL = 0.687) does not intersect with zero, which confirms full mediation effects ($\beta = 0.380$, $p < 0.001$ f2 = 0.059) and supports H5.

4.2.2. Moderation Results

To investigate the moderating impact of environmental uncertainty and disruption orientation, the study has compared the path coefficients of high- and low-risk categories to see whether there are any significant changes [113]. Thus, the moderating effects are checked and evaluated using bootstrapping tests with 5000 samples in SmartPLS 3 to determine the component weights, T-statistics, and significance of the path coefficients. Table 4 and Figures 2 and 3 display the results, which reveal that environmental uncertainty moderates the relationship between supply chain resilience and operational performance in a negative direction ($\beta = -0.380$, $p < 0.012$). Thus, the higher the environmental uncertainty, the lower the relationship between supply chain resilience and operational performance. This supports H6. On the other hand, the results have reported that disruption orientation ($\beta = 0.108$, $p > 0.472$) does not affect the relationship between supply chain resilience and operational performance. Consequently, H7 is not supported.

**5. Discussions and Conclusions**

This study is grounded on the DCV to gauge the impact of five dynamic capabilities on hotel FSCs resilience and to highlight resilience influences on operational performance. The findings of this study provide credence to our hypotheses and mostly offer a different perspective on the determinants that influence supply chain resilience. Most empirical models have failed to account for this complex aspect of supply chain resilience. Our results show that the value of dynamic capabilities in relation to supply chain resilience and operational performance outcomes in emerging economies, such as Egypt, is even more significant. Given that the research sample is deemed to be a true reflection of the various hotels in Egypt, it is important to note that these hotels, which fall under the four- and five-star categories, must meet the minimum benchmarks set for global hotel rankings.

Consequently, the findings derived from this study can potentially be extrapolated to a broader context of the international hospitality industry, extending beyond the borders of Egypt. Several factors play a role in the generalization of the findings of the present study. Firstly, it is worth noting that all hotels within the designated study areas (i.e., Greater Cairo, the Northern Coast, Sharm El Sheikh, Dahab, El Gouna, Marsa Alam, Hurghada, and Ras Sidr) were given an equal chance of being chosen, ensuring a fair representation of the overall hotel industry. Additionally, the sample size utilized in the study was carefully determined to effectively capture the characteristics of the broader study population from a statistical standpoint. The study's findings also postulate that the operational performance of hotel FSCs during disruptions can be maintained or even enhanced through resilience gained from the adaptive dynamic capabilities that hotels can provide in collaboration with their partners. Our findings generally confirm the direct and indirect relation between hotels' dynamic capabilities (in total) and the operational performance in the hospitality sector. These results are in line with previous research, e.g., [42,76], which has argued that dynamic capabilities have a significant influence on performance. Additionally, it is found that SC resilience statistically mediates the relationship between hotels' dynamic capabilities (in total) and operational performance. Conformingly, Alkalha et al. [89] have stated that supply chain resilience strongly mediates the relationship between dynamic capabilities in terms of absorptive capacity and operational performance. Thus, hotel operators and/or owners should enhance their resource capabilities to achieve higher performance through the resilience of their food supply chains. The more the hotel's dynamic capabilities are, the higher the hotel FSC resilience will occur, and this will, in turn, lead to better hotel FSC performance.

Furthermore, the results affirm that collaboration improves supply chain resilience. These results agree with the findings of [3,40,51,114]. As a result, we argue that hotels' supply chain risk mitigation can be influenced by their collaborative efforts, which are influenced by factors such as sharing of information, effective communication with partners, the development of mutually beneficial bodies of knowledge, and cooperative working relationships. Hence, it is reasonable to assume collaboration is a prerequisite for supply chain resilience. This finding is aligned with the arguments of prior studies that to establish supply chain resilience, enterprises must merge and adjust their resources, processes, and capabilities [20].

As expected, integration significantly influences supply chain resilience. This result reinforces previous findings that supply chain integration enables firms to respond to unforeseen disruptions and thus increases supply chain resilience. These findings resonate with those revealed by [24,60,62]. Therefore, hotels must cooperate and exchange data with partners to respond to changing market demands by integrating internal and external information systems to face difficulties when disruptions occur. As a result, strategic management theory in a post-pandemic and non-ergodic world is considered more resilient. Although integration facilitates the exchange of information, hotels should be wary of its potential drawbacks. Al Naimi et al. [59] argue that supply chain integration increases risk and forces all stakeholders to share those risks.

In addition, agility has a significantly positive influence on supply chain resilience. This aligns with the findings of [5,41,114]. This also indicates that improvements in hotel FSCs' adaptability, transparency, and velocity may be needed to mitigate and adapt quickly to severe environmental and operational uncertainty.

Surprisingly, and in contrast to earlier findings [9,18,27,77], responsiveness is not found to be significantly influencing supply chain resilience in the hotel context ($\beta = 0.018$, $p > 0.329$). Bearing in mind that the majority of respondents are decision-makers, these findings are important to shed more light on the response behavior of this group during times of risk and turmoil. So, our results show that the ability to respond quickly to changing customer and supplier needs is not a top priority for the hotel sector in Egypt. This might be attributed to a plausible problem with the hotels' internal logistics systems or inadequate support from relevant actors that prevents hotels from receiving a prompt

response during disruptions. Our findings also contradict those of [27,55]. It is evident that resource reconfiguration positively influences hotel FSCs resilience during disruptions that cause a shortage of resources. This adds to what Ambulkar et al. [103] and Parker & Ameen [72] have reported about the value of supply chain resource reconfiguration.

Our results confirm the hypothesized relationship between supply chain resilience and operational performance. This finding aligns with previous studies that state that resilience can improve supply chain performance [20,115]. The ability to bounce back quickly from setbacks is crucial in highly unpredictable situations. So, hotels need to be resilient and react quickly to changes in the market and fluctuations in the availability of key supplies. Besides, as previously indicated, resilience can only be achieved with full integration with the partners. Hence, the more hotel FSCs resilience is, the less negative effects of disruption loss of hotel FSCs occur. This can, in turn, lead to better hotel FSC performance.

Our findings further confirm the moderating effect of environmental uncertainty on the relationship between resilience practices and supply chain operational performance. This finding can inspire hotels in Egypt to implement environmental scanning practices to decipher early warning signals. Silvestre [116] has argued that uncertainty in developing countries' environments hinders supply chains from benefiting from the accumulation of dynamic capabilities and thus impedes performance improvement. Surprisingly, our findings reveal that disruption orientation does not have a moderating effect on the relationship between resilience practices and supply chain operational performance. Our results are inconsistent with those found in the literature that emphasizes the significance of gaining insight from prior disruptions [103,117].

## 6. Implications

### 6.1. Theoretical Implications

This study sheds light on the impact of dynamic capabilities on the resilience of hotel FSCs by developing a model based on the DCV theory and filling in the gaps in the previous literature. Our model characterizes the resilience of hotel FSCs in terms of collaboration, integration, agility, responsiveness, and reconfiguration. This model also develops and validates the impact of hotel FSCs resilience in relation to operational performance. Our findings add to the existing literature on hotel FSC management in developing countries like Egypt. Furthermore, this study paves the way for hotels to reinforce their FSCs during disruptions, which is especially crucial considering the volatility of the country's economy. The validated study model fills in the gaps with regard to what has been known about hotel FSCs. It also measures how well these chains, especially those in developing countries, can adapt to current and future challenges. Our perspective that resilience is a crucial interim step between supply chain dynamic capabilities and operational performance is a new contribution to the literature. The empirical findings confirm that taking advantage of opportunities during disruptions requires a resilient mindset [5], and this way of thinking must extend to the supply chain level. In addition, this study has inserted relational moderators that may play an important role in developing the resilience of hotel FSCs. Thus, we have broadened the scope of the research on dynamic capacities and highlighted that these capabilities can be used as a beneficial approach by hotels to ensure food supply chain resilience.

### 6.2. Practical Implications

Our findings should aid hotel managers in planning for disruptions. The results show collaboration, integration, agility, and reconfiguration are essential for developing supply chain resilience. The study has proven that collaboration and integration with supply chain partners develop supply chain resilience. Thus, managers must acknowledge the significance of supply chain dynamic capabilities for enhancing operational performance to successfully adapt to evolving surroundings. Therefore, integration with partners is no longer an option. Hotels should enhance proactive and reactive activities with their key partners through mutual innovation, joint problem-solving, simultaneous decisions,

pooled resources, and aligned incentives. Hotels and their partners need to turn to the full implementation of digital technology to build an integrated database.

Information sharing between partners in day-to-day practices enhances supply chain visibility. Visibility in hotel food supply chains permits the early detection of supply disruptions. In a collaborative and integrated setting, information sharing is the adhesive that holds the whole thing together and motivates the productive actions necessary to achieve the set goals [118]. To achieve visibility and supply chain resilience, hotels need to consider the frequency, direction, and manner of shared information [119]. Internal integration can be limited to data like orders, forecasts, shipping information, upcoming disruptions, market trends, and maintenance schedules. Similarly, hotel supply chain management decision-makers should focus on resources to capitalize on hotels' ability related to disruption orientation. For example, with trained staff and recent technologies, such as artificial intelligence, robotics, simulation, and advanced predictive analytics, hotels can understand the dynamics of previous disruptions more deeply and make predictions for the next wave.

Hotels should focus on human-centred leadership and continuous improvement to become more agile and respond quickly to real threats to the supply chain operation. Finally, hotels that desire to improve the operational performance of their supply chains need to continually assess the design of their supply chains and use proactive and reactive approaches to enhance resilience. Managers can also use the study scale as a diagnostic tool to determine which dynamic capabilities are dominant and have a remarkable effect on managing supply chain vulnerabilities. To deal with unexpected events, hotel supply chains often discover, integrate, and organize dynamic capabilities. Hotels do not only learn from the results of these activities (whether positive or negative) but also from the process itself, which improves their ability to assess and prepare for future responses. The current uncertainty has made it difficult for many hotels in Egypt to maintain their standard supply chain operations. Therefore, we believe that it is time for hotels in Egypt to evaluate their operations, even if past results have shown success in obtaining the targeted goals at this level.

## 7. Limitations and Directions for Future Research

Although we have sought to impose accuracy in the procedures of this study, some limitations must be considered when interpreting the study results. The fact that we have used data from only four five-star hotels in Egypt is the main limitation of our study. Caution should be exercised when extrapolating the study's findings to other categories of hotels. Additionally, the study has been conducted in the service sector, particularly the hospitality sector. Thus, future research could replicate the study in other sectors (i.e., industrial and service). The data were also collected in the aftermath of the COVID-19 epidemic and the early days of the Russian-Ukrainian conflict. This means that the causal relationships established in this study apply to the unique situation of hotels that respond to these crises. The causal relationship may have developed rapidly since the completion of data collection. Finally, no data have been collected to determine the hotels' financial performance during the disruption. It would be interesting for researchers to find out more about the relationship between financial indicators and the resilience of supply chains.

**Author Contributions:** Conceptualization, M.A.K., O.M.A., S.S. and O.A.; Methodology, M.H.A.A.-A. and O.A.; Software, M.A.K., O.M.A., S.S. and O.A.; Validation, M.A.K., O.M.A., S.S., M.H.A.A.-A. and O.A.; Formal analysis, M.A.K., S.S., M.H.A.A.-A. and O.A.; Investigation, O.M.A. and O.A.; Resources, S.S. and O.A.; Data curation, M.A.K. and O.A.; Writing – original draft, M.A.K., O.M.A., S.S. and O.A.; Writing – review & editing, M.A.K., S.S., M.H.A.A.-A. and O.A.; Visualization, M.A.K.; Project administration, M.A.K.; Funding acquisition, M.H.A.A.-A. All authors have read and agreed to the published version of the manuscript.

**Funding:** This research received no external funding.

**Institutional Review Board Statement:** Not applicable.

**Informed Consent Statement:** Not applicable.

**Data Availability Statement:** Not available.

**Acknowledgments:** We thank our families for their support and the participants in the study.

**Conflicts of Interest:** The authors declare no conflict of interest.

## Appendix A

**Table A1.** Item Loadings and Descriptive Results.

| Construct/Item | Loadings | Mean | SD | Kurtosis | Skewness |
|---|---|---|---|---|---|
| *Agility Capability (AGI)* | | | | | |
| AG1: The hotel swiftly adjusts services and/or goods to new customer requirements | 0.60 | 4.169 | 0.792 | −0.037 | −0.693 |
| AG2: The hotel responds fast to market developments | 0.81 | 4.131 | 0.653 | −0.676 | −0.141 |
| AG3: The hotel responds swiftly to substantial demand rises and declines | 0.82 | 3.725 | 0.866 | −1.441 | 0.566 |
| AG4: The hotel adapts its product portfolio as per market requirement | 0.55 | 3.606 | 0.526 | −1.162 | −0.045 |
| AG5: The hotel reacts to changes in competition strategy faster than our competitors | 0.82 | 3.85 | 0.831 | −1.503 | 0.289 |
| *Collaboration Capability (COL)* | | | | | |
| COL1: The hotel has a partnership arrangement | 0.89 | 4.062 | 0.772 | −0.339 | −0.438 |
| COL2: The hotel actively participates in collective decision-making with its partners | 0.86 | 4.431 | 0.892 | 1.015 | −1.392 |
| COL3: The hotel actively interacts with partners in collaborative problem-solving | 0.84 | 3.569 | 1.105 | 1.111 | −1.496 |
| COL4: The hotel and its partners have an excellent working relationship | 0.83 | 3.987 | 1.374 | −0.158 | −0.999 |
| COL5: In partnership with our partners, the hotel sets strategic goals | 0.70 | 3.75 | 1.067 | −0.216 | −0.421 |
| *Disruption Orientation (DIO)* | | | | | |
| DO1: At all times, the hotel is concerned about prospective supply chain disruptions | 0.66 | 3.231 | 0.464 | 0.791 | 1.082 |
| DO2: The hotel recognizes that supply chain disruptions are always looming | 0.78 | 3.013 | 0.962 | −1.672 | 0.017 |
| DO3: The hotel thinks a lot about how a supply chain disruption could be avoided | 0.83 | 3.319 | 0.529 | 1.063 | 1.409 |
| DO4: Supply chain disruptions are extensively analyzed | 0.93 | 3.438 | 1.197 | −1.424 | −0.105 |
| *Environmental Uncertainty (ENU)* | | | | | |
| EU1: Hotel demand varies significantly from week to week | 0.91 | 3.556 | 0.927 | −0.564 | 0.523 |
| EU2: The actions of the hotel's competitors are unpredictable | 0.76 | 3.175 | 1.11 | −0.966 | 0.617 |
| EU3: The needs and preferences of the customers are erratic | 0.91 | 3.206 | 1.141 | −1.04 | 0.454 |
| EU4: It is necessary to make major changes in the production processes | 0.85 | 3.294 | 1.076 | −1.005 | 0.515 |
| EU5: The hotel's menus become obsolete at a rapid rate | 0.85 | 3.331 | 1.094 | −0.965 | 0.352 |
| EU6: The industry's technology is quickly evolving | 0.88 | 3.269 | 1.094 | −1.116 | 0.432 |

**Table A1.** *Cont.*

| Construct/Item | Loadings | Mean | SD | Kurtosis | Skewness |
|---|---|---|---|---|---|
| *Integration Capability (INT)* | | | | | |
| INT2: The hotel can communicate customized data externally with key suppliers | 0.83 | 3.062 | 0.772 | 1.866 | −1.178 |
| INT4: The hotel eliminates partner recurrence | 0.81 | 3.281 | 0.8 | 0.839 | −1.072 |
| INT5: The hotel ensures data consistency with partners | 0.82 | 3.237 | 0.87 | 0.678 | −1.116 |
| INT6: The hotel collaborates with partners to predict and arrange the activities | 0.85 | 3.15 | 1.062 | 0.065 | 0.107 |
| INT7: The hotel logistics ITs are extended to include more integrated applications | 0.87 | 3.1 | 1.097 | 0.015 | 0.115 |
| INT8: The hotel logistics ITs capture and maintain timely data | 0.83 | 3.138 | 1.335 | −1.398 | 0.016 |
| *Operational Performance (OPP)* | | | | | |
| OP1: The efficacy of our SC in fulfilling its goals | 0.79 | 3.244 | 0.92 | 0.679 | −0.359 |
| OP2: The ability of our SC to adapt to changes in market demand | 0.78 | 3.256 | 0.917 | 0.736 | −0.385 |
| OP5: Shortening the lead time | 0.92 | 3.594 | 1.027 | 1.845 | −1.708 |
| OP6: Reduction of overhead expenses | 0.84 | 3.312 | 1.119 | 0.21 | −1.129 |
| *Reconfiguring Capability (REC)* | | | | | |
| REC1: The hotel can reconfigure supply chain resources to create new productive assets | 0.87 | 3.275 | 0.487 | 1.557 | 0.186 |
| REC2: The hotel can align capabilities to match the current supply chain requirements | 0.85 | 3.125 | 0.82 | −0.883 | −0.099 |
| REC3: The hotel can efficiently integrate and mix current resources into unique combinations | 0.95 | 3.2 | 1.418 | −1.328 | 0.053 |
| *Responsiveness Capability (RES)* | | | | | |
| RES1: The hotel adapts quickly to changing customer demands | 0.92 | 4.025 | 0.88 | 3.879 | −1.605 |
| RES3: The hotel reacts more rapidly and effectively to rivals' quality strategies | 0.88 | 3.525 | 0.928 | −0.509 | 0.707 |
| RES4: The hotel responds swiftly to changing supplier scope | 0.89 | 3.831 | 1.108 | 0.153 | −0.94 |
| *Supply Chain Resilience (SCR)* | | | | | |
| SCR1: The hotel SC can promptly respond to unexpected disruptions by quickly restoring its product flow | 0.89 | 3.369 | 1.197 | −0.433 | −0.105 |
| SCR2: In the event of a disruption, the hotel SC can swiftly return to its original state | 0.92 | 3.406 | 1.142 | −0.375 | −0.059 |
| SCR3: After being disrupted, the hotel SC might evolve into a more desired state | 0.91 | 3.619 | 1.134 | 0.168 | −0.713 |
| SCR4: The hotel supply chain is well prepared to deal with the outcomes of disruptions | 0.90 | 3.194 | 1.272 | −1.022 | 0.108 |
| SCR5: The hotel SC can keep structure and operation under control even during disruptions | 0.88 | 3.45 | 1.396 | −1.187 | −0.481 |
| SCR6: Disruptions (unexpected events) may be turned into meaningful learning opportunities for the hotel | 0.83 | 3.513 | 0.981 | −0.353 | 0.346 |

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
