# Peer review of "Dynamic Capabilities Influence on the Operational Performance of Hotel Food Supply Chains: A Mediation-Moderation Model"

_sustainability, doi:10.3390/su151813562_

Round 1

Reviewer 1 Report

The paper studies the impact of dynamic capabilities on the resilience and operational performance of supply chains. A focused case study is taken as example, regarding the hotel food supply chain. The Dynamics Capabilities View (DCV) is employed, together with a questionnaire to hotel managers and structural equation modeling.

The topic is of interest. The problem is well defined. The research methodology is well described and can be replicated.

To improve the paper:

1. the questionnaire must be added as an appendix

2. the capabilities of the instruments (DCV model, questionnaire, equations) to be generalized (e.g., applied to any other industrial sector or supply chain) must be mentioned in an independent section.

Reviewer 2 Report

This paper studied dynamic capabilities’ influence to resilience and operational performance of hotel food supply chain. The topic is of practical significance. However there are some major concerns:

1.   If the research is about resilience influence to operational performance, I can understand. Why is resilience and operational performance are considered at the same time as dependent variables, this research design is difficult to understand.

2.   Although definition of dynamic capabilities is introduced, how it is disaggregated into collaboration, integration, agility etc is not clear.

3.   How to measure resilience and operational performance needs to be discussed.

4.   Theoretical framework is lack of support. Figure 1 is in poor quality. Some words are not shown. There is no resilience introduced in theoretical framework. Hypothesis is not complied with theoretical framework. I would suggest resilience as a mediating variable.

5.   There is lack of discussion of sample size, simple representativeness.

6.   It is advisable to update the reference list with the most recent and relevant high quality literature, such as:

Amir Karbassi Yazdi, Amir Mehdiabadi, Peter Fernandes Wanke, Nazli Monajemzadeh, Henrique Luiz Correa & Yong Tan (2023) Developing supply chain resilience: a robust multi-criteria decision analysis method for transportation service provider selection under uncertainty, International Journal of Management Science and Engineering Management,18:1, 51-64

Sharmistha Halder Jana (2022) Application of expected value and chance constraint on uncertain supply chain model with cost, risk and visibility for COVID-19 pandemic, International Journal of Management Science and Engineering Management, 17:1, 10-24

Sharfuddin Ahmed Khan, Syed Mehmood Hassan, Simonov Kusi-Sarpong, Muhammad Shujaat Mubarik & Sana Fatima (2022) Designing an integrated decision support system to link supply chain processes performance with time to market, International Journal of Management Science and Engineering Management, 17:1, 66-78

Language is undstandable.

Reviewer 3 Report

1. The title may be modified as: Effect of dynamic capabilities on the resilience and operational performance under the moderating role of  environmental uncertainty and disruption orientation: Evidence from hotel food supply chains

2. The authors should have mentioned the impact of disruption orientation in the abstract section. 

3. There is a repetition of the numbering of sections: Introduction and Theoretical Background and Hypotheses development

4. Figure 1: Some texts (in the boxes) are not readable. 

5. Sample size adequacy is not checked

6. The result section needs to be more concisely presented. Please relate the findings with the novelty of the paper.

7. The conclusion section should highlight the future directions in a more explicit way.

Minor changes are required.

Round 2

Reviewer 2 Report

This paper has been improved, especially the Theoretical Framework and Hypotheses. I have no objection to the acceptance of the paper.

Authors may consider the generalizability of the research results. Since data only collected in Egyptian hotel, can the research results be generalized to other background?

I recommend authors to polish the whole paper. As there are some obvious expression issues. Mixed tenses are used. Some literature review is in present tense, some is in past tense. There are some very long and wordy sentences. 

Author Response

Dear reviewer 

Thank you so much for your valuable comments. All comments have been considered and the whole manuscript has been revisited.